# Scaling at quantum phase transitions above the upper critical dimension

Anja Langheld[1][*], Jan A. Koziol[1][◇], Patrick Adelhardt[1][†], Sebastian C. Kapfer[1][‡] and Kai P. Schmidt[1][◦]

**1** Department of Physics, Staudtstraße 7, Friedrich-Alexander-Universität Erlangen-Nürnberg, Germany

[*] anja.langheld@fau.de
[◇] jan.koziol@fau.de
[†] patrick.adelhardt@fau.de
[‡] sebastian.kapfer@fau.de
[◦] kai.phillip.schmidt@fau.de

July 12, 2022

## Abstract

The hyperscaling relation and standard finite-size scaling (FSS) are known to break down above the upper critical dimension due to dangerous irrelevant variables. We establish a coherent formalism for FSS at quantum phase transitions above the upper critical dimension following the recently introduced Q-FSS formalism for thermal phase transitions. Contrary to long-standing belief, the correlation sector is affected by dangerous irrelevant variables. The presented formalism recovers a generalized hyperscaling relation and FSS form. Using this new FSS formalism, we determine the full set of critical exponents for the long-range transverse-field Ising chain in all criticality regimes ranging from the nearest-neighbor to the long-range mean field regime. For the same model, we also explicitly confirm the effect of dangerous irrelevant variables on the characteristic length scale.

# 1   Introduction

Finite-size scaling (FSS) provides an important tool for extracting critical properties from finite systems. It allows one to extrapolate to the thermodynamic limit by exploiting the generalized homogeneity of observables provided by renormalization group (RG) theory [1]. In spite of being widely used for several decades already, FSS has been insufficiently understood above the upper critical dimension $d_{uc}$ for a long time. In the context of RG theory, FSS [2] has been proven below the upper critical dimension [3] and the breakdown of which for $d > d_{uc}$ has been identified to be due to the presence of dangerous irrelevant variables (DIV) in the free energy sector [4, 5]. Even though irrelevant variables flow to zero under successive renormalization, a DIV cannot be set to zero as the free energy density $f$ is singular in this limit [4]. In contrast, the correlation sector was thought to be unaffected by DIV for a long time [4, 6–9]. This breakdown of FSS is often associated with the breakdown of hyperscaling

$$(d + z)\nu = 2 - \alpha \tag{1}$$

which is clearly violated above the upper critical dimension, where the critical exponents do not depend on the dimension anymore [6]. Historically, in an attempt to fix FSS above the upper critical dimension for thermal phase transitions, an additional characteristic length scale, the so-called thermodynamic length scale, was introduced [5, 7, 10]. Although this approach is capable of producing correct results in many frameworks [7, 10–17], the theory does not capture the full picture in a coherent way [9, 18, 19] as it neither explains the anomalous scaling of the characteristic length scale $\xi$ [20] nor the anomalous decay of the correlation function [14, 19, 21–23] above the upper critical dimension. Moreover, the theory is based on the disputable claim that the correlation sector is unaffected by the DIV [6–9, 19].

Recently, the topic of FSS above the upper critical dimension has been revisited for thermal phase transitions [9, 18, 19, 24]. By relaxing the claim that the characteristic length scale $\xi$, that diverges at the critical point, is bound by the linear system size and allowing the correlation sector to be affected by DIV, they derived a coherent picture of FSS above the upper critical dimension for classical systems, called Q-FSS [9,19,24], and derived a generalized hyperscaling relation [18,24]. We aim to transfer Q-FSS to quantum phase transitions, which we refer to as quantum Q-FSS, pointing out the necessary steps while stressing the differences and connection to the classical counterpart.

Besides the need for a fundamental understanding of FSS, finite-size effects inevitably play a role in experiments and numerical simulations of finite systems. For systems which can be modelled by a theory above the upper critical dimension, there is a demand for a generalized FSS

formalism which is also applicable in the regime $d > d_{uc}$. In particular, the upper critical dimension becomes accessible in low dimensions for systems with unfrustrated long-range interactions as these lower the upper critical dimension with respect to the short-range counterparts [25]. These long-range interactions have received a lot of interest in quantum systems lately as they exhibit remarkable quantum critical properties [26–30].

Algebraically decaying interactions are present in dipolar systems such as Rydberg atoms [31] and in systems of trapped ions [29,32–42], where the decay exponent of the long-range couplings $\sim |\mathbf{x}|^{-d-\sigma}$ can be continuously tuned [29,39–42]. It shall be explicitly stressed that those experimental platforms realize systems with a mesoscopic number of well controlled entities in contrast to solid-state bulk systems.

With vast progress being made in the experimental realization of long-range interacting quantum systems, the demand for a theoretical understanding of this long-range regime has increased. In particular, the long-range transverse-field Ising model (LRTFIM) with algebraically decaying Ising couplings has become a paradigmatic model to study the effects of long-range interactions [25,43–54]. For a ferromagnetic coupling, those give rise to a continuum of universality classes for small decay exponents $\sigma$ going over to a mean field regime for even smaller $\sigma$ [47,48,51–54]. In this regime, simulations of finite systems cannot be extrapolated to the thermodynamic limit within the standard formalism [52]. Even methods operating in the thermodynamic limit potentially need to make use of a generalized hyperscaling relation that is also valid above the upper critical dimension for extracting the whole set of critical exponents in all criticality regimes.

Following the spirit of classical Q-FSS [18], we provide a quantum Q-FSS formalism unifying the scaling predictions from RG with the criticality known from mean field calculations. In the course of this, we will derive - similar to the classical case [18] - a generalized hyperscaling relation which is also valid above the upper critical dimension. We will validate our theory by applying it to numerical data of the one-dimensional ferromagnetic LRTFIM obtained by quantum Monte Carlo (QMC) simulations and high-order series expansions using perturbative continuous unitary transformations (pCUT) and demonstrate the extraction of the full set of critical exponents using quantum Q-FSS.

The paper is divided into three main parts: Sec. 2 covers the theoretical part of this paper in which we derive quantum Q-FSS, while in Sec. 3 we validate quantum Q-FSS based on a numerical study of the one-dimensional ferromagnetic LRTFIM. We give a conclusion of this work in Sec. 4. In detail, we start Sec. 2 with a brief description of the necessary scaling framework in the vicinity of a continuous quantum phase transition. Adressing the scaling of finite systems, we first consider the well-behaved szenario below the upper critical dimension in Sec. 2.1. Quantum Q-FSS is then gradually derived in Sec. 2.2 starting with a treatment of DIV similar to the classical case which leads to a modified scaling of observables. In Sec. 2.2.1, a necessary argument that fixes the scaling with the linear system size is transferred to the quantum case. As the main results, a generalized hyperscaling relation and quantum Q-FSS are presented in Sec. 2.2.2 and Sec. 2.2.3 respectively. In Sec. 2.2.4 we draw the connection to classical Q-FSS and discuss different perspectives on the modified scaling above the upper critical dimension. In Sec. 3 we verify quantum Q-FSS and the generalized hyperscaling relation derived in Sec. 2 by numerically calculating the full set of critical exponents of the one-dimensional LRTFIM in its three criticality regimes by means of QMC and high-order series expansions. After introducing the LRTFIM in Sec. 3.1, we briefly introduce the two numerical methods together with the observables we measure in Sec. 3.2. In Sec. 3.3 we present the critical exponents directly extracted by the two methods respectively as well as the full set of critical exponents calculated by using scaling relations including the generalized hyperscaling relation. Furthermore, we provide numerical evidence that the correlation sector is

affected by DIV by studying the FSS of the characteristic length scale.

## 2 Scaling at continuous quantum phase transitions

We consider a system close to a second-order quantum phase transition with three relevant parameters $r, H, T$ vanishing at the critical point. With $r \sim \lambda - \lambda_c$ being the distance of the control parameter $\lambda$ to the critical control parameter value $\lambda_c$, $H$ denoting the symmetry-breaking field coupling to the order parameter $m$ and $T$ being the temperature. Without loss of generality, the system is in its symmetry-broken phase for $r < 0$ and in the symmetric phase for $r > 0$. At the quantum critical point $r, H, T = 0$, the characteristic length scale $\xi$ diverges and the physical quantities exhibit singular behavior in the form of power laws, which get characterized by critical exponents:

$$\chi_r = \frac{\partial^2 f}{\partial r^2} \sim |r|^{-\alpha} \qquad m(r \to 0^-) \sim |r|^{\beta} \qquad m \sim |H|^{1/\delta} \qquad \chi = \frac{\partial^2 f}{\partial H^2}\bigg|_{H=0} \sim |r|^{-\gamma} \tag{2}$$

$$G(\mathbf{q}, \omega = 0) \sim |\mathbf{q}|^{-(2-\eta)} \qquad \xi \sim |r|^{-\nu} \qquad \xi_\tau \sim |r|^{-z\nu}.$$

Widom proposed the generalized homogeneity of those functions close to a critical point [55, 56], which was later understood within the framework of RG [57]. Extending this generalized homogeneity to finite systems by including the inverse linear system size $1/L$ as an additional relevant parameter is the basis of FSS [1, 58]. Close to the critical point, the singular part of the free energy density $f$ and characteristic length $\xi$ asymptotically become generalized homogeneous functions (GHF) [6, 57–60]

$$f(r, H, T, L^{-1}, u) = b^{-(d+z)} f(b^{y_r} r, b^{y_H} H, b^z T, b L^{-1}, b^{y_u} u) \tag{3}$$

$$\xi(r, H, T, L^{-1}, u) = b \xi(b^{y_r} r, b^{y_H} H, b^z T, b L^{-1}, b^{y_u} u) \tag{4}$$

depending on the couplings $r, H, T, u$ and the inverse system length $L^{-1}$ with the respective scaling dimensions $y_r, y_H, z > 0$, $y_L = 1$, and $y_u < 0$ governing the linearized RG flow with spatial rescaling factor $b > 1$ around the RG fixed point, at which all couplings vanish by definition. All of those couplings are relevant except for $u$ which denotes the leading irrelevant coupling [61,62]. Other irrelevant couplings with scaling dimensions smaller than $y_u$ have already been set to zero as $f$ is assumed to be analytic in these parameters. The scaling power $y_r$ of the control parameter $r$ is related to the critical exponent $\nu$ by $y_r = \nu^{-1}$ [62]. The generalized homogeneity of the other observables in Eq. (2) follows from the generalized homogeneity of $f$ (for details see Ref. [59]), e. g., the generalized homogeneity of the magnetization

$$m(r, H, T, L^{-1}, u) = b^{-(d+z)+y_r} m(b^{y_r} r, b^{y_H} H, b^z T, b L^{-1}, b^{y_u} u) \tag{5}$$

follows from taking the derivative of $f$ with respect to $H$.

### 2.1 Scaling below the upper critical dimension

Below the upper critical dimension, $f$ is an analytic function in $u$ and one can safely set $u = 0$ in the homogeneity laws, dropping the dependence on $u$ in Eq. (3)

$$f(r, H, T, L^{-1}) = b^{-(d+z)} f(b^{y_r} r, b^{y_H} H, b^z T, b L^{-1}) \tag{6}$$

as well as in the homogeneity laws for the other observables as it was already done for the other irrelevant couplings.

By probing the singular behavior of the respective GHFs at the critical point approaching it along one of the principal axes [59], one can relate the scaling dimensions of the relevant variables with the critical exponents

$$\alpha = -\frac{d+z-2y_r}{y_r}, \quad \beta = \frac{d+z-y_H}{y_r}, \quad \delta = \frac{y_H}{d+z-y_H}, \quad \gamma = -\frac{d+z-2y_H}{y_r}, \tag{7}$$

from which some of the scaling relations, including the hyperscaling relation $2 - \alpha = (d+z)\nu$, can be extracted when additionally using $y_r^{-1} = \nu$. Expressing the scaling dimensions in terms of the critical exponents, the homogeneity law for an observable $\mathcal{O}$ with bulk divergence $\mathcal{O}(r, 0, 0, 0) \sim |r|^\omega$ is given by

$$\mathcal{O}(r, H, T, L^{-1}) = b^{-\omega y_r} \mathcal{O}(b^{y_r} r, b^{y_H} H, b^z T, b L^{-1}) \tag{8}$$

$$= b^{-\omega/\nu} \mathcal{O}(b^{1/\nu} r, b^{(\beta+\gamma)/\nu} H, b^z T, b L^{-1}), \tag{9}$$

where Eq. (7) was used to express $y_H = (\beta + \gamma) y_r$ in terms of critical exponents. From that, FSS is readily obtained by setting $b = L$, thereby fixing the last entry to $b L^{-1} = 1$,

$$\mathcal{O}(r, H, T, L^{-1}) = L^{-\omega/\nu} \Psi(L^{1/\nu} r, L^{(\beta+\gamma)/\nu} H, L^z T) \tag{10}$$

with $\Psi$ being the universal scaling function of the observable $\mathcal{O}$.

However, this FSS only holds for $d < d_{\mathrm{uc}}$ since $u$ is a DIV above the upper critical dimension, meaning that $f(r, H, T, L^{-1}, u)$ is singular at $u = 0$, which renders the homogeneity Eq. (6) for $f(r, H, T, L^{-1}, u = 0)$ meaningless. We will now explicitly consider the case in which $u$ is a DIV.

## 2.2 Scaling above the upper critical dimension

As $u$ is a DIV above the upper critical dimension, it cannot be dropped in the homogeneity relations. Instead, we assume for the singular part of the free energy density for small $u$ [6]

$$f(r, H, T, L^{-1}, u) = u^{p_{(d+z)}} \bar{f}(u^{p_r} r, u^{p_H} H, u^{p_T} T, u^{p_L} L^{-1}), \tag{11}$$

so that the dependence on $u$ can be absorbed into the other variables up to a global power $p_{(d+z)}$ of $u$. This implies a modified scaling for the free energy density [6]

$$f(r, H, T, L^{-1}) = b^{-(d+z)^*} f(b^{y_r^*} r, b^{y_H^*} H, b^{z^*} T, b^{y_L^*} L^{-1}) \tag{12}$$

$$= L^{-(d+z)^*/y_L^*} \mathcal{F}(L^{y_r^*/y_L^*} r, L^{y_H^*/y_L^*} H, L^{z^*/y_L^*} T) \tag{13}$$

by defining the modified scaling powers [6]

$$(d+z)^* = (d+z) - p_{(d+z)} y_u, \qquad \begin{aligned} y_r^* &= y_r + p_r y_u, & y_H^* &= y_H + p_H y_u, \\ z^* &= z + p_z y_u, & y_L^* &= 1 + p_L y_u. \end{aligned} \tag{14}$$

In contrast to the classical case [6], where $y_L^*$ was implicitly set to $y_L^* = y_L = 1$ by the authors, we allow $y_L^*$ to be distinct from $y_L$. Fixing $y_L^*$ is possible because the scaling powers of a GHF are only determined up to a common non-zero factor [59] such that there is a freedom to set one non-zero scaling power to an arbitrary non-zero value. We keep the derivation general and postpone the

discussion of specific choices for the absolute values of the modified scaling powers to Sec. 2.2.4 as it has no impact on the derivation of the generalized hyperscaling relation (see Sec. 2.2.2) or Q-FSS (see Sec. 2.2.3).

For a long time, the correlation sector was thought to be unaffected by DIV [6–8]. However, classical Q-FSS [9, 18] showed that the correlation sector needs a reexamination as well [9]. In analogy to classical Q-FSS [9,18], we therefore allow the characteristic length scale $\xi$ to be affected by DIV analogous to the free energy sector. This results in the modified scaling [6, 18, 24]

$$\xi(r, H, T, L^{-1}) = b^{-y_\xi^*}\xi(b^{y_r^*}r, b^{y_H^*}H, b^{z^*}T, b^{y_L^*}L^{-1}) \tag{15}$$

$$= L^\varrho \Xi(L^{y_r^*/y_L^*}r, L^{y_H^*/y_L^*}H, L^{z^*/y_L^*}T) \tag{16}$$

with $y_\xi^* = -1 - p_\xi y_u = -y_r^*/y_r$ in order to reproduce the correct bulk singularity $\xi \sim |r|^{-\nu}$ and defining a "pseudocritical exponent"[1] $\varrho$ ("koppa"), which is related to the critical exponent $\nu$ by

$$\varrho = \frac{y_r^*}{y_r y_L^*} = \nu \frac{y_r^*}{y_L^*}. \tag{17}$$

Analogous to the case below the upper critial dimension, the modified scaling powers can be related to the critical exponents by comparing the singularities of the thermodynamic functions in terms of critical exponents with the singularities of the respective GHFs in terms of the modified scaling powers. This leads to very similar equations

$$\alpha = -\frac{(d+z)^* - 2y_r^*}{y_r^*}, \tag{18}$$

$$\beta = \frac{(d+z)^* - y_H^*}{y_r^*}, \tag{19}$$

$$\delta = \frac{y_H^*}{(d+z)^* - y_H^*}, \tag{20}$$

$$\gamma = -\frac{(d+z)^* - 2y_H^*}{y_r^*}, \tag{21}$$

which fix the quotients of the modified scaling powers to

$$y_r^* = \frac{(d+z)^*}{2}, \qquad\qquad y_H^* = \frac{3(d+z)^*}{4} \tag{22}$$

when inserting the mean field critical exponents $\alpha = 0$ and $\delta = 3$.

We want to note that it is to be expected that only the ratios of the modified scaling powers can be fixed as the scaling powers of a GHF can be rescaled with a common non-zero factor without altering the GHF [59]. Since Eqs. (18) – (21) were extracted from the scaling of infinite systems, they only relate the modified scaling powers determining the bulk scaling. However, in order to extend the scaling to finite systems, the modified scaling power $y_L^*$ needs to be connected to the bulk scaling powers $y_r^*, y_H^*$, and $(d+z)^*$ as well. For this, one has to consider the scaling of *finite* systems as it was done for the classical counterpart in Ref. [6]. After finding this last missing ratio intrinsic to the GHF, the homogeneity relations are completely determined as the absolute values of the scaling powers are meaningless.

---

[1]This new exponent $\varrho$ was sometimes called critical exponent [24] as well as pseudocritical exponent [63], but often simply referred to as "exponent" in the past. We choose to call $\varrho$ a pseudocritical exponent due to its very similar definition to bulk critical exponents as the ratio of scaling powers defining the power-law behavior of a thermodynamic quantity in one parameter along its respective principal axis. In contrast to critical exponents, $\varrho$ defines a power law with respect to a finite linear system size $L$ and is therefore not defined at criticality, at which $L$ is infinite.

### 2.2.1 Linking bulk scaling to finite systems

Comparing the finite-size scaling of the order-parameter susceptibility with the scaling of its modified GHF structure, Binder *et al.* [6] argued for the classical case that $d^* = d$ given that $y_L^* = 1$. By transferring this argument to quantum systems, we will derive a non-trivial relation for $(d + z)^*$. Unlike in the original argument [6], we will continue to leave $y_L^*$ unspecified to keep the derivation general. Like Binder *et al.* [6], we consider the susceptibility in a finite system at $H = T = 0$ which, for a quantum system, is given by an infinite integral over imaginary time

$$\chi_L = L^d \int_0^\infty \langle m(\tau)m(0)\rangle_L \, d\tau \, , \tag{23}$$

introducing the short-hand notation $\mathcal{O}_L = \mathcal{O}(r, H = 0, T = 0, L^{-1})$. In contrast to the classical case, where $m$ commutes with $\mathcal{H}$ and the susceptibility reduces to $\chi_L = L^d \beta \langle m^2\rangle_L$, we need to take the scaling due to the imaginary-time integral into account. As the correlations are expected to decay exponentially in imaginary time $\langle m(\tau)m(0)\rangle_L \sim e^{-\Delta_L \tau}\langle m^2\rangle_L$ with the finite-size energy gap $\Delta_L \sim \xi_{\tau,L}^{-1}$, the integration gives

$$\chi_L \sim L^d \left\langle m^2\right\rangle_L \Delta_L^{-1} \, . \tag{24}$$

For sufficiently large systems, $\langle m^2\rangle_L$ and $\Delta_L$ take on their bulk values $\langle m^2\rangle_\infty \sim |r|^{2\beta}$ and $\Delta_\infty \sim |r|^{z\nu}$ and the susceptibility scales as

$$\chi_L \sim L^d |r|^{2\beta}|r|^{-z\nu} \tag{25}$$

close to the critical point $r = 0$. This scaling has to be compatible with the GHF structure of the susceptibility

$$\chi(r, H, T, L^{-1}) = L^{[-(d+z)^* + 2y_H^*]/y_L^*} \mathcal{X}(L^{y_r^*/y_L^*}r, L^{y_H^*/y_L^*}H, L^{z^*/y_L^*}T) \, , \tag{26}$$

that follows from taking the second derivative of Eq. (13) with respect to $H$. We therefore require the scaling function $\mathcal{X}$ for large $L$ to scale as

$$\lim_{x \to \pm\infty} \mathcal{X}(x, 0, 0) \sim |x|^{2\beta - z\nu} \tag{27}$$

to reproduce the correct bulk singularity in $|r|$ and further demand

$$\frac{-(d+z)^* + 2y_H^* + (2\beta - z\nu)y_r^*}{y_L^*} = d \tag{28}$$

in order to match the scaling in $L$. Using Eq. (19) and $\nu = 1/y_r$, we eliminate the critical exponents $\beta$ and $\nu$ in Eq. (28) and obtain

$$(d+z)^* = y_L^* d + \frac{y_r^*}{y_r}z \, , \tag{29}$$

which relates the modified scaling power $y_L^*$ of the inverse linear system size $L^{-1}$ with the modified bulk scaling powers. This is an important step towards deriving a FSS form above the upper critical dimension as the FSS is governed by the ratio of $y_L^*$ with the other modified scaling powers (see, e. g., Eqs. (13) and (26)). With this, all ratios of modified scaling powers are known, fixing them

up to a common non-zero factor that can be chosen freely (see Sec. 2.2.4). This global factor does not alter our results which we are about to discuss, starting with a generalized hyperscaling relation. However, one can already identify two meaningful choices from Eq. (29): For the choice $y_L^* = 1$ that was also taken for the classical counterpart [6, 18], $(d+z)^* = d+\digamma z$ and the scaling of imaginary time and temperature seems to be modified by a factor of $\digamma$. On the other hand, when leaving the scaling power $y_r^* = y_r$ unmodified, $(d+z)^* = d/\digamma + z$ and the temperature scaling seems unmodified while the spatial dimension is reduced by a factor of $\digamma$.

### 2.2.2 Generalized hyperscaling relation

Hyperscaling is commonly said to break down above the upper critical dimension due to the emergence of DIV [5]. We obtain a generalized hyperscaling relation from Eq. (18), which relates the critical exponent $\alpha$ with the modified scaling powers. Inserting Eq. (29) into Eq. (18) yields

$$2 - \alpha = \left( \frac{d}{\digamma} + z \right) \nu, \tag{30}$$

where it was additionally used that $\digamma^{-1} = y_L^* y_r / y_r^*$ and $\nu = y_r^{-1}$. This already shows that rather the spatial dimensions behave different instead of the imaginary-time dimension because $z\nu$ is unaltered while $d\nu \to d\nu/\digamma$ with respect to the usual hyperscaling relation.

This generalized hyperscaling relation also yields a way to determine the new pseudocritical exponent $\digamma$. The regular hyperscaling relation is still valid at $d = d_{uc}$ and relates the mean field values for $\alpha$ and $\nu$ via $2-\alpha = (d_{uc}+z)\nu$. As the mean field critical exponents also hold for $d > d_{uc}$, this relation remains valid above the upper critical dimension. Comparing it with the generalized hyperscaling relation Eq. (30) gives

$$\digamma = \frac{d}{d_{uc}} \qquad \text{for } d > d_{uc} \tag{31}$$

for the new pseudocritical exponent. The ratio $\digamma/\nu = y_r^*/y_L^*$ will be of particular importance in the quantum Q-FSS form describing the finite-size scaling of observables in finite systems above the upper critical dimension. In the classical case [9, 18, 19, 24], a system with spatial dimension $D > D_{uc}$ and an upper critical dimension $D_{uc}$ has a pseudo-critical exponent $\digamma_{cl} = D/D_{uc}$ that governs the scaling $\xi_L \sim L^{\digamma_{cl}}$ at the classical critical point. Considering the quantum-classical mapping and demanding that the exponent $\digamma$ of a quantum system should coincide with the $\digamma_{cl}$ of its classical analogue, the generalization of Q-FSS to the quantum case would yield $\digamma = (d+z)/(d_{uc}+z)$ which clearly differs from Eq. (31) for any non-zero $z$ and $d \neq d_{uc}$. This apparent contradiction will be resolved in Sec. 2.2.4 by taking a closer look on the quantum-classical correspondence.

### 2.2.3 Quantum Q-Finite-size scaling

As an important result, FSS above the upper critical dimension is derived. Like standard FSS [2], it predicts the rounding of physical quantities in finite systems with respect to the bulk behavior. Analogous to the case $d < d_{uc}$, for an observable $\mathcal{O}$ with bulk divergence $\mathcal{O}(r,0,0,0) \sim |r|^\omega$ the

GHF structure is given by [2]

$$\mathcal{O}(r, H, T, L^{-1}) = b^{-\omega y_r^*} \mathcal{O}(b^{y_r^*} r, b^{(\beta+\gamma)y_r^*} H, b^{z^*} T, b^{y_L^*} L^{-1}) \tag{32}$$

$$= L^{-\omega \mathqoppa/\nu} \Psi(L^{\mathqoppa/\nu} r, L^{(\beta+\gamma)\mathqoppa/\nu} H, L^{z^*/y_L^*} T). \tag{33}$$

This also holds for the special case of the characteristic length scale $\xi$ with $\omega = -\nu$ (see Eq. (15)). As a result of $\mathqoppa \neq 1$, the characteristic length scale of a finite system does not scale linearly with the linear system size at the critical point, but with

$$\xi_L \sim L^{\mathqoppa}. \tag{34}$$

This is an important, non-trivial consequence of Q-FSS as the characteristic length scale was formerly thought to be bound by the linear system size [6] until this claim was relaxed by classical Q-FSS [18, 24].

   With Eq. (33) and $\mathqoppa = d/d_{uc}$ for $d > d_{uc}$, the rounding of finite systems with respect to the bulk behavior is characterized in terms of a universal scaling function $\Psi$ and the critical exponents. It can be used to extract critical exponents in a mean field regime from finite systems [52, 64]. In Ref. [52], this allowed us to benchmark the employed algorithm in the numerically challenging regime of long-range interactions.

   In complete analogy to the classical case [19], the definition of $\mathqoppa$ can be extended to $d < d_{uc}$ by setting

$$\mathqoppa = \max\left(1, \frac{d}{d_{uc}}\right) \tag{35}$$

such that the FSS form Eq. (33) holds below as well as above the upper critical dimension. For $d < d_{uc}$, the exponent $\mathqoppa = 1$ recovers the standard FSS forms and the linear scaling of the characteristic length scale $\xi_L \sim L$.

### 2.2.4 Comparing classical and quantum Q-FSS

As for now, the classical and quantum Q-FSS appear analogous, but there are some important differences to note. The classical pseudocritical exponent $\mathqoppa_{cl} = D/D_{uc}$ is also given by the ratio of the dimension $D$ of the classical system with respect to its upper critical dimension $D_{uc}$. However, this means that the exponent $\mathqoppa$ of a quantum system differs from the exponent $\mathqoppa_{cl}$ of its classical $D = (d+1)$-dimensional analogue, e.g., for the 4d nearest-neighbor transverse-field Ising model (TFIM), $\mathqoppa = 4/3$ (see App. A for numerical evidence), while for its classical analogue, the 5d classical Ising model, $\mathqoppa_{cl} = 5/4$ [18, 20, 24]. The important difference is that the quantum system is only finite in $d$ dimensions with the imaginary time dimension - constituting the additional classical dimension - being infinite at zero temperature. The $(d+1)$-dimensional classical analogue of a $d$-dimensional finite quantum system at zero temperature therefore has a geometry $L^d \times \infty$ with $d = D - 1$ finite dimensions and one infinite dimension.[3]

---

[2] We leave the scaling power $z^*$ of the temperature undetermined in this scaling law. It governs the FSS of the finite-size gap with $\Delta_L \sim L^{z^*}$. Based on the modified scaling power $(d + z)^*$ (see Eq. (29)), $z$ appears to be modified as $z \to \frac{y_r^*}{y_r} z$ and we therefore conjecture that $z^* = \frac{y_r^*}{y_r} z$, because the scaling power $(d + z)^*$ is the scaling power of the Euclidean spacetime volume and, similarily, we have a scaling power $y_L^*$ for the spatial dimension which is also reflected in $d \to y_L^* d$ in Eq. (29).

[3] In practice, it is sufficient to have a finite inverse temperature $\beta \gg \xi_{L,\tau} \sim \Delta_L^{-1}$ for which the system does not feel its finite extent in imaginary time. For larger temperatures we expect a crossover to classical Q-FSS.

Table 1: Modified scaling powers for the (LR)TFIM when choosing $p_L = 0$ or $p_r = 0$ respectively. The choice $p_L$ corresponds to leaving the scaling power of the inverse system size invariant, while the choice $p_r$ leaves the scaling power of the characteristic length scale invariant.

|  |  | $p_L = 0$ |  | $p_r = 0$ |  |
|---|---|---|---|---|---|
| $y_L^*$ | $y_L + p_L y_u$ | $1$ | $p_L = 0$ | $1/\varrho$ | $p_L = -\frac{1}{d}$ |
| $(d+z)^*$ | $d+z - p_{(d+z)} y_u$ | $d + \varrho z$ | $p_d = \frac{1}{3}$ | $d_{\mathrm{uc}} + z$ | $p_d = -1$ |
| $y_r^*$ | $y_r + p_r y_u$ | $\frac{2}{3} d$ | $p_r = -\frac{2}{3}$ | $y_r$ | $p_r = 0$ |
| $y_H^*$ | $y_H + p_H y_u$ | $\frac{3}{4}(d + \varrho z)$ | $p_H = -\frac{1}{2}$ | $\frac{3}{4}(d_{\mathrm{uc}} + z)$ | $p_H = \frac{1}{2}$ |
| $z^*$ | $z + p_z y_u$ | $\varrho z$ | $p_z = -\frac{1}{3}$ | $z$ | $p_z = 0$ |

Table 2: Modified scaling powers for the (long-range) Ising model when choosing $p_L = 0$ or $p_r = 0$ respectively. The choice $p_L$ corresponds to leaving the scaling power of the inverse system size invariant, while the choice $p_r$ leaves the scaling power of the characteristic length scale invariant.

|  |  | $p_L = 0$ |  | $p_r = 0$ |  |
|---|---|---|---|---|---|
| $y_L^*$ | $y_L + p_L y_u$ | $1$ | $p_L = 0$ | $1/\varrho_{\mathrm{cl}}$ | $p_L = -\frac{1}{D}$ |
| $d^*$ | $d - p_d y_u$ | $d$ | $p_d = 0$ | $D_{\mathrm{uc}}$ | $p_d = -1$ |
| $y_r^*$ | $y_r + p_r y_u$ | $\frac{1}{2} d$ | $p_r = -\frac{1}{2}$ | $y_r$ | $p_r = 0$ |
| $y_H^*$ | $y_H + p_H y_u$ | $\frac{3}{4} d$ | $p_H = -\frac{1}{4}$ | $\frac{3}{4} D_{\mathrm{uc}}$ | $p_H = \frac{1}{2}$ |

This is supported by a study of Brézin [3]. He showed for the classical spherical model that in a geometry $L^{D-1} \times \infty$, the correlation length at the critical point $T = T_c$ for $D > D_{\mathrm{uc}}$ scales as [3]

$$\xi_L \sim L^{(D-1)/(D_{\mathrm{uc}}-1)} \tag{36}$$

with the linear system size $L$. This result was argued to also hold for finite $N$ in the $N$-vector model [3], which includes the Ising model for $N = 1$. The 5d classical Ising model with geometry $L^4 \times \infty$ therefore has a $\varrho = (D-1)/(D_{\mathrm{uc}} - 1) = 4/3$ which is in line with the quantum-classical mapping.

The link between classical and quantum Q-FSS is also visible in a certain choice for the modified scaling powers. Up to now, their absolute values remained unspecified as only their ratios enter into the generalized hyperscaling relation Eq. (30) and Q-FSS form Eq. (33). As the characteristic length scale $\xi$ does no longer scale linearly with $L$ for $d > d_{\mathrm{uc}}$, one of these length scales inevitably scales with a scaling power that is unusual for a length scale in RG. The two choices we consider are given by leaving $y_L^* = y_L = 1$ invariant, meaning $p_L = 0$, or by leaving the scaling power $y_\xi$ of $\xi$ invariant by setting $y_\xi^* = -y_r^*/y_r = -1$, meaning $p_\xi = p_r = 0$. The resulting modified scaling powers are compared for the (LR)TFIM in Tab. 1 and for the classical (long-range) Ising model in Tab. 2.

The first point to note is that, when the scaling power of the characteristic length scale remains the one of a length scale in RG (see the right columns in Tab. 1 and Tab. 2), the modifying $p$-factors are equivalent for the quantum and classical model and the total modified bulk scaling powers are equivalent for models in terms of the quantum-classical mapping, which is consistent with the equivalence of criticality in terms of the quantum-classical mapping. Only the scaling power for $L^{-1}$ differs due to the difference in $\varrho$ and $\varrho_{\mathrm{cl}}$. On the other hand, when choosing the linear system

size $L$ to scale as an ordinary length scale in RG, neither the modifying $p$-factors nor the modified scaling powers $y_r^*$ coincide. Only $y_L^* = 1$ in both cases coincides by construction.

Even though the absolute values of the scaling powers are not important for Q-FSS, we prefer the picture in which $\xi$ retains its scaling power $y_\xi = -1$ as a length scale in RG. Not only because the modifications of the quantum and classical systems are equivalent by virtue of the quantum-classical mapping, but also because the length scale $\xi$ is also apparent in bulk systems. Here, the scaling does not depend on $L^{-1} = 0$ and the choice of modified scaling powers retains meaningful. Moreover, for $y_\xi^* = -1$, the bulk scaling powers are independent of the dimension $d$ just as the criticality is independent of the dimension for $d > d_{\text{uc}}$.

# 3 Application and verification of quantum Q-FSS

So far, we derived a coherent quantum Q-FSS theory with a generalized hyperscaling relation and FSS forms to extract critical exponents from simulations of finite systems. In this section we verify this formalism by applying it to the LRTFIM on the linear chain. We compute - using two independent numerical methods - a full set of critical exponents below and above the upper critical dimension. On the one hand, we simulate finite systems using stochastic series expansion (SSE) [65–68] quantum Monte Carlo (QMC) to demonstrate the application of the FSS scaling forms Eq. (33). On the other hand, we use perturbative continuous unitary transformations (pCUT) [69,70], a series expansion method in the thermodynamic limit, to demonstrate the application of the generalized hyperscaling relation Eq. (30). Further, we explicitly calculate the characteristic length scale $\xi_L$ on finite systems with SSE QMC in order to demonstrate its scaling with the pseudocritical exponent $\mathcal{P}$ and support our claim that the correlation sector is affected by DIV. The raw data used in this work as well as the extracted scaling dimensions of the respective calculated quantities and their conversion to critical exponents is provided in Ref. [71].

## 3.1 Long-range transverse-field Ising chain

We consider the ferromagnetic LRTFIM on the linear chain. The Hamiltonian is given by

$$\mathcal{H}^{\text{LRTFIM}} = \frac{J}{2} \sum_{i \neq j} \frac{1}{|i-j|^{1+\sigma}} \sigma_i^z \, \sigma_j^z - h \sum_j \sigma_j^x \, , \tag{37}$$

with Pauli matrices $\sigma_i^{x/z}$ describing spins 1/2 located on lattice sites $i$. The transverse field is tuned by the parameter $h > 0$ while the ferromagnetic Ising coupling is tuned by the parameter $J < 0$. The positive parameter $(1 + \sigma)$ governs the decay of the Ising coupling constants, from a nearest-neighbor model for $\sigma = \infty$ to an all-to-all coupling for $\sigma = -1$.

The model exhibits a continuous quantum phase transition between a field-polarized and a symmetry-broken ferromagnetic phase for all $\sigma > 0$. Its upper critical dimension is lowered with decreasing decay parameter $\sigma$ and the universality class of the quantum critical point varies as a function of this decay parameter. One may identify three different regimes, namely a regime with the well-known 2d Ising criticality for large $\sigma$, a long-range Gaussian regime with mean field criticality for small $\sigma$ as well as an intermediate regime with continuously varying critical exponents connecting the two limiting regimes. Those regimes and in particular their boundaries can be understood by means of a field-theoretical analysis of the critical point [25,53,72] considering

the one-component quantum rotor action [25]

$$\mathcal{A} = \frac{1}{2} \int_{q,\omega} (\tilde{g}\omega^2 + aq^\sigma + bq^2 + r)\tilde{\phi}_{q,i\omega}\tilde{\phi}_{-q,-i\omega} + u \int_{x,\tau} \phi_{x,\tau}^4 \tag{38}$$

with $a, b > 0$ and $r, u$ being the mass and coupling term [25]. For $\sigma \geq 2$ the leading term in $q$ recovers the nearest-neighbor $\phi^4$ Ising action with the $(d+1)$-dimensional Ising criticality [62]. A detailed analysis of the RG flow of the kinetic sector [53] suggests that this Ising criticality holds for even smaller decay exponents $\sigma > 2 - \eta_{\mathrm{SR}}$ with $\eta_{\mathrm{SR}}$ being the anomalous dimension of the respective short-range model. By a scaling analysis of the Gaussian theory for $\sigma < 2$, it is possible to derive the long-range mean field critical exponents [25]

$$\gamma = 1, \qquad \nu = \frac{1}{\sigma}, \qquad \eta = 2 - \sigma, \qquad z = \frac{\sigma}{2}. \tag{39}$$

By inserting these exponents into the hyperscaling relation, the upper critical dimension $d_{\mathrm{uc}}(\sigma)$ can be derived with [25]

$$2 - \alpha = \nu(d + z) \qquad \xrightarrow[\nu=\frac{1}{\sigma},\ z=\frac{\sigma}{2}]{\alpha=0} \qquad d_{\mathrm{uc}} = \frac{3\sigma}{2}. \tag{40}$$

For the linear chain with $d = 1$, this results in a regime of mean field criticality, in which $d > d_{\mathrm{uc}}$, for

$$\sigma < \frac{2}{3}. \tag{41}$$

In this regime, standard FSS is not applicable and the critical exponents can be extracted from finite systems only by means of quantum Q-FSS (see Eq. (33)). Inserting the value for the upper critical dimension and $d = 1$ into the expression of the pseudocritical exponent $\digamma$ yields

$$\digamma = \max\left(1, \frac{2}{3\sigma}\right) = \begin{cases} 1 & \text{for } \sigma \geq 2/3 \\ \frac{2}{3\sigma} & \text{for } \sigma < 2/3. \end{cases} \tag{42}$$

In the FSS analysis we will encounter several combinations of critical exponents, for which the analytic values are known in the limiting cases of the mean field regime with $\sigma < 2/3$ and of the short-range regime with $\sigma > 2 - \eta_{\mathrm{SR}}$. These combinations of exponents are given in Tab. 3.

Table 3: Analytical values in the limiting regimes for the exponents that are directly accessible by the methods presented in Sec. 3.2.

| Regime | $\frac{\nu}{\digamma}$ | $\beta\frac{\digamma}{\nu}$ | $\gamma\frac{\digamma}{\nu}$ | $z\nu$ | $(2-z-\eta)\nu$ | $\alpha$ |
|---|---|---|---|---|---|---|
| $\sigma < 2/3$ | $\frac{\sigma^{-1}}{2/(3\sigma)} = \frac{3}{2}$ | $\frac{1}{2}\cdot\frac{2}{3} = \frac{1}{3}$ | $1\cdot\frac{2}{3} = \frac{2}{3}$ | $\frac{\sigma}{2}\cdot\frac{1}{\sigma} = \frac{1}{2}$ | $(\sigma-\frac{\sigma}{2})/\sigma = \frac{1}{2}$ | $0$ |
| $\sigma > 2-\eta_{\mathrm{SR}}$ | $1$ | $\frac{1}{2}$ | $1$ | $1$ | $2-1-\frac{1}{4} = \frac{3}{4}$ | $0$ |

## 3.2 Numerical methods and observables

We study the one-dimensional LRTFIM with two different methods to validate Q-FSS for quantum systems. While SSE operates on finite systems, pCUT operates in the thermodynamic limit. For the former, we exploit the quantum Q-FSS form Eq. (33) to extract critical exponents from simulations of finite systems. In both cases, we utilize the generalized hyperscaling relation Eq. (29) in addition to other scaling relations to extract the full set of critical exponents.

### 3.2.1 Stochastic series expansion

We use the SSE QMC approach introduced by A. Sandvik to sample the transverse-field Ising model with arbitrary Ising interactions $J_{ij}$ on arbitrary graphs [65–68]. The SSE approach is based on a high temperature expansion of the partition function

$$Z = \text{Tr}\{e^{-\beta\mathcal{H}}\} = \sum_{n=0}^{\infty} \sum_{\{|\alpha\rangle\}} \frac{(-\beta)^n}{n!} \langle\alpha|\mathcal{H}^n|\alpha\rangle \tag{43}$$

with the idea to extend the configuration space in imaginary time by using an adequate decomposition $\mathcal{H} = -\sum_i \mathcal{H}_i$ and rewriting [65–68]

$$\mathcal{H}^n = (-1)^n \sum_{\{S_n\}} \prod_{p=1}^{n} \mathcal{H}_i \tag{44}$$

as sequences $S_n$ of the operators $\mathcal{H}_i$. This extended configuration space is then sampled by a Markov chain. We will not go into the details of the algorithm as we follow precisely the scheme described in Ref. [52].

The SSE method is a finite-temperature Quantum Monte Carlo technique. To obtain ground-state results, the temperature of the simulation needs to be sufficiently low for the contribution of excited states to the averaged observables to be negligible. A systematic approach to ensure convergence in temperature within the statistical Monte Carlo error was described in Ref. [52]. By the application of this scheme, all observables are measured at effectively zero temperature.

As in Ref. [52], we determine the mean squared magnetization $\langle m^2\rangle_L$ for a set of transverse fields $h$ and system sizes $L$, where

$$m = \frac{1}{L} \sum_i \sigma_i^z \tag{45}$$

is the order parameter of the investigated quantum phase transition. In this work, we additionally calculate the order-parameter susceptibility

$$\chi_L = L \int_0^{\beta} \langle m(\tau)m(0)\rangle_L \, d\tau \tag{46}$$

using the algorithm of Sandvik and Kurkijärvi [65].

By performing a data collapse of $\langle m^2\rangle_L$ and $\chi_L$ using the scaling predictions from Q-FSS (see Eq. (33))

$$\begin{aligned}
\langle m^2\rangle_L(r) &= L^{-2\beta\digamma/\nu}\mathcal{M}(L^{\digamma/\nu}r), \\
\chi_L(r) &= L^{\gamma\digamma/\nu}\mathcal{X}(L^{\digamma/\nu}r)
\end{aligned} \tag{47}$$

for vanishing longitudinal field $H = 0$ and effectively vanishing temperature $T = 0$ with $r \sim h - h_{\text{c}}$ and universal scaling functions $\mathcal{M}$ and $\mathcal{X}$, we extract the exponents $\nu/\digamma$, $\beta\digamma/\nu$ and $\gamma\digamma/\nu$. The data collapse was performed as described in Ref. [52].

In addition to the common critical exponents, we also measure the pseudocritical exponent $\digamma$ in the mean field regime by performing a data collapse of the finite-size characteristic length scale with scaling

$$\xi_L(r) = L^{\digamma}\Xi(L^{\digamma/\nu}r) \tag{48}$$

according to Q-FSS. As $\nu = \sigma^{-1}$ is known from the Gaussian theory, the only free parameters in the fit are $\varsigma$, $h_c$, and the scaling function $\Xi$.

For measuring the characteristic length scale $\xi$, at which the correlations switch to their long-distance behavior [62, 73], we consider the order-parameter correlation function

$$G_L(i - j, \omega = 0) = \left. \frac{\partial \langle \sigma_i^z \rangle_L}{\partial H_j} \right|_{H_j = 0}$$
$$= \int_0^\beta \left\langle \sigma_i^z(\tau) \sigma_j^z(0) \right\rangle_L \mathrm{d}\tau \tag{49}$$

with $H_j$ being a local longitudinal field coupling to spin $\sigma_j^z$ at lattice site $j$. The correlation function Eq. (49) is the zero-frequency component of the Fourier transform of the imaginary-time correlation function

$$G_L(i - j, \tau) = \left\langle \sigma_i^z(\tau) \sigma_j^z(0) \right\rangle_L , \tag{50}$$

which also contains information on the dynamics of the system. We are not interested in any dynamical properties in this study and will therefore only use the zero-frequency correlation function in Eq. (49) as well as the equal-time correlation function

$$G_L(i - j, \tau = 0) = \left\langle \sigma_i^z \sigma_j^z \right\rangle_L . \tag{51}$$

Using the Fourier transform of the correlation functions for $\tau = 0$ and $\omega = 0$, we extract the characteristic length scale in the long-range mean field regime, where the criticality is described by a Gaussian field theory. The correlation function in Fourier space is given by the propagator of the long-range Gaussian field theory [25, 63]

$$\tilde{G}(q, \omega) \sim \frac{1}{aq^\sigma + \tilde{g}\omega^2 + m^2} \tag{52}$$

with $m$ the characteristic energy scale, which in terms of the coupling $r$ is given by $m^2 \sim |r|$. This yields the zero-frequency and equal-time correlation functions [74]

$$\tilde{G}(q, \omega = 0) \sim \frac{1}{aq^\sigma + m^2} , \tag{53}$$

$$\tilde{G}(q, \tau = 0) \sim \frac{1}{2\sqrt{\tilde{g}}\sqrt{aq^\sigma + m^2}} . \tag{54}$$

For a finite system, the definition of a characteristic length scale in terms of the correlation function is ambiguous [75]. There are several definitions for $\xi_L$ which will converge to $\xi_\infty$ for $L \to \infty$ [75]. For long-range systems, finding a suitable definition for the characteristic length is even more difficult, as the correlation function does not exhibit the usual exponential decay of gapped systems but decays algebraically even away from the critical point [73]. Common definitions that are tailored for correlation lengths, which specify the exponential decay of a correlation function at long distances, such as the second moment

$$\xi_\infty^{(2)} = \sqrt{\frac{1}{2d} \frac{\int |\mathbf{x}|^2 G(\mathbf{x}) \, \mathrm{d}\mathbf{x}}{\int G(\mathbf{x}) \, \mathrm{d}\mathbf{x}}} , \tag{55}$$

therefore might yield $\xi_\infty^{(2)} = \infty$ in an infinite system not only at the critical point $r = 0$, but also for $r \neq 0$ [76].

We will instead consider the definition [63]

$$\xi_L^{(\text{LR}\omega)} = \frac{1}{q_{\min}} \left[ \frac{\tilde{G}_L(0, \omega = 0) - \tilde{G}_L(q_{\min}, \omega = 0)}{\tilde{G}_L(q_{\min}, \omega = 0)} \right]^{1/\sigma} \tag{56}$$

with $q_{\min} = 2\pi/L$ the smallest wavevector fitting on the finite lattice. By inserting Eq. (53)

$$\xi_L^{(\text{LR}\omega)} = \frac{1}{q_{\min}} \left[ \frac{aq_{\min}^\sigma + m_L^2}{m_L^2} - 1 \right]^{1/\sigma} = a^{1/\sigma} m_L^{-2/\sigma} \tag{57}$$

the momentum dependency cancels.[4] In case of the equal-time correlation function, we use the square of $\tilde{G}(q, \tau = 0)$ in order to remove the square-root in Eq. (54) which yields a slightly modified formula

$$\xi_L^{(\text{LR}\tau)} = \frac{1}{q_{\min}} \left[ \frac{\tilde{G}_L^2(0, \tau = 0) - \tilde{G}_L^2(q_{\min}, \tau = 0)}{\tilde{G}_L^2(q_{\min}, \tau = 0)} \right]^{1/\sigma} = a^{1/\sigma} m_L^{-2/\sigma} \tag{58}$$

for the same quantity. In the limit $L \to \infty$, the estimates for the characteristic length exhibit the correct singularity

$$\xi_\infty^{(\text{LR})} = a^{1/\sigma} m_\infty^{-2/\sigma} \sim |r|^{-1/\sigma} \sim |r|^{-\nu}. \tag{59}$$

### 3.2.2 Perturbative continuous unitary transformation

We also use high-order series expansions in the thermodynamic limit employing the method of perturbative continuous unitary transformations (pCUT) [69, 70] to extract critical exponents. The pCUT method comes with the prerequisite that the Hamiltonian must take the form

$$\mathcal{H} = \mathcal{H}_0 + \mathcal{V} = E_0 + \mathcal{Q} + \sum_{m=-N}^{N} T_m \tag{60}$$

with the unperturbed Hamiltonian $\mathcal{H}_0 = E_0 + \mathcal{Q}$ where $E_0$ is the unperturbed ground-state energy, $\mathcal{Q}$ counts the number of quasi-particles, and the perturbation $\mathcal{V}$ which must decompose into a sum of operators $T_m$ that change the system's energy by $m$ quanta such that $[\mathcal{Q}, T_m] = mT_m$. The fundamental idea of pCUT is to transform the original Hamilitonian $\mathcal{H}$ perturbatively order by order into an effective quasiparticle-conserving Hamiltonian $\mathcal{H}_{\text{eff}}$ mapping the complicated many-body problem to an easier effective few-body problem with $[\mathcal{Q}, \mathcal{H}_{\text{eff}}] = 0$. The effective Hamiltonian $\mathcal{H}_{\text{eff}}$ contains products of $T_m$ operators with exact rational coefficients and is independent of the exact form of $\mathcal{H}$ [69]. Similarly, observables $\mathcal{O}$ can be mapped to effective observables $\mathcal{O}_{\text{eff}}$ resulting in an expression analogous to the Hamiltonian. However, the quasiparticle-conserving property is lost [70].

These model-independent expressions of the effective Hamiltonian and observables come at the cost of a second, model-dependent step, where the Hamiltonian must be normal-ordered. This is usually done by a full-graph decomposition applying the effective Hamiltonian or observable to

---

[4]It is important that we use a zero-momentum quantity to observe the anomalous scaling of the characteristic length scale because only zero-momentum quantities are affected by DIV for periodic boundary conditions. A detailed analysis of the role of Fourier modes can be found in Ref. [9].

topologically distinct finite clusters exploiting the linked-cluster theorem which states that only linked processes have an overall contribution to cluster-additive quantities [77]. For long-range interactions, linked-cluster expansions are only feasible using white graphs [47,48,77]. This means that additional information is encoded into a multivariable polynomial during the application of pCUT while edge colors (colors would correspond to distinct distances between interacting sites) on graphs are ignored in the topological classification of white graphs reducing the amount of contributing graphs to a finite number [47,48,51,77].

After employing the pCUT method on white graphs, the resulting contributions must be embedded on the infinite chain to determine the contributions in the thermodynamic limit. For each realization of a graph on the lattice, every variable of the multivariable polynomial must be replaced by the proper coupling strength $\sim |i - j|^{-1-\sigma}$. Due to the hard-core constraint that graph vertices must not overlap, the embedding procedure is equivalent to evaluating infinite nested sums. As the sums become quite tedious, it is advantageous to evaluate them using Markov chain Monte Carlo integration [47]. For a generic quantity $\kappa$, the embedding problem can be written as

$$\kappa = \sum_n c_n^\kappa \lambda^n, \qquad c_n^\kappa = \sum_{N=2}^{n+1} S[f_N], \tag{61}$$

where $S[\cdot]$ is the Monte Carlo sum over all possible configurations and $f_N$ contains all graph contributions from graphs with $N$ vertices since the sums are identical for a given number of graph vertices.

To extract the critical point and exponents beyond the radius of convergence of the pure perturbative series we use DlogPadé extrapolations as described in Ref. [78]. We define the Padé extrapolant

$$P[L,M]_\mathcal{D} = \frac{P_L(\lambda)}{Q_m(\lambda)} = \frac{p_0 + p_1\lambda + \cdots + p_L\lambda^L}{1 + q_1\lambda + \cdots + q_M\lambda^M} \tag{62}$$

with $p_i, q_i \in \mathbb{R}$ of the logarithmic derivative $\mathcal{D}(\lambda) = \frac{\mathrm{d}}{\mathrm{d}\lambda}\ln(\kappa)$ with a physical quantity $\kappa$ which is given as a perturbative series up to order $r$. The extrapolant is determined such that its Taylor expansion up to order $r - 1 = L + M$ must recover the series of $\mathcal{D}(\lambda)$. The DlogPadé extrapolant of $\kappa$ is then defined as

$$\mathrm{d}P[L,M]_\kappa = \exp\left(\int_0^\lambda P[L,M]_\mathcal{D} \, \mathrm{d}\lambda'\right). \tag{63}$$

Given a dominant power-law behavior $\kappa \sim |\lambda - \lambda_c|^{-\theta}$, an estimate for the critical point $\lambda_c$ can be determined by analyzing the poles of $P[L,M]_\mathcal{D}$ as well as for the exponent calculating the residuum $\mathrm{Res}\, P[L,M]_\mathcal{D}|_{\lambda=\lambda_c}$ at the physical pole. If $\lambda_c$ is known, we can define biased Dlog-Padés by the Padé extrapolant $P[L,M]_{\theta^*}$ of $\theta^* = (\lambda_c - \lambda)\frac{\mathrm{d}}{\mathrm{d}\lambda}\ln(\kappa)$. As before, the critical exponent $\theta$ can be determined by the residuum of $P[L,M]_{\theta^*}$ at the physical pole. If further the exponent $\hat{\theta}$ of the multiplicative logartihmic correction to the dominant power-law is known $\theta^* = (\lambda_c - \lambda)\frac{\mathrm{d}}{\mathrm{d}\lambda}\ln(\kappa) + \hat{\theta}/\ln(1 - \lambda/\lambda_c)$ can be used.

Now, turning to the LRTFIM in Eq. (37), we perform high-oder series expansions about the high-field limit $h \gg J$. The reference state $|\mathrm{ref}\rangle$ in the unperturbed limit is the fully polarized state $|\rightarrow \cdots \rightarrow\rangle$ with local spin-flips $|\leftarrow_j\rangle$ at arbitrary positions $j$ as elementary excitations. By rescaling the energy spectrum of the Hamiltonian with $1/2h$ and using the Matsubara-Matsuda transformation [79] to express the Hamiltonian in terms of hard-core boson operators $b_j^{(\dagger)}$, we

arrive at

$$\mathcal{H} = E_0 + \mathcal{Q} + \sum_{i<j} \lambda(i-j)\left( b_i^\dagger b_j + b_i^\dagger b_j^\dagger + H.c. \right) \tag{64}$$

with the unperturbed ground-state energy $E_0 = -N/2$ with $N$ the number of sites, $\mathcal{Q} = \sum_i b_i^\dagger b_i$ counting the number of quasiparticles (QPs), and the coupling term $\lambda(i-j) = J/2h \times |i-j|^{-1-\sigma}$. After normal-ordering and Fourier transformation, the effective 1QP Hamiltonian becomes

$$\tilde{\mathcal{H}}_{\text{eff}}^{\text{1QP}} = \bar{E}_0 + \sum_k \omega(k) b_k^\dagger b_k, \tag{65}$$

where $\bar{E}_0$ is the ground-state energy and $\omega(k)$ the 1QP dispersion. Note that we do not consider multi-QP properties in this work. With the Hamiltoninan in this form, the control-parameter susceptibility and the 1QP excitation gap can be directly determined by

$$\chi_r = \frac{\mathrm{d}^2 \bar{E}_0}{\mathrm{d}\lambda^2}, \qquad \Delta = \min_k \omega(k). \tag{66}$$

Further, we choose to calculate the 1QP static spectral weight as an observable. Starting from the usual definition of the static structure factor and exploiting that it decomposes into a sum of spectral weights, we arrive at

$$\mathcal{S}^{z,\text{1QP}}(k) = \left| \langle k | \sigma_{\text{eff},k}^z | \text{ref} \rangle \right|^2 = |s(k)|^2 \tag{67}$$

with $\sigma_{\text{eff},k}^{z,\text{1QP}} = s(k)(b_k^\dagger + b_k)$ being the effective Pauli $z$-operator in second quantization restricted to the 1QP channel. Finally, we note the well-known critical behavior

$$\chi_r \sim |\lambda - \lambda_c|^{-\alpha}, \qquad \Delta \sim |\lambda - \lambda_c|^{z\nu}, \qquad \mathcal{S}^{z,\text{1QP}}(k_{\text{crit}}) \sim |\lambda - \lambda_c|^{-(2-z-\eta)\nu} \tag{68}$$

of the control-parameter susceptibility, the 1QP excitation gap, and the 1QP static spectral weight. Here, we calculated the ground-state energy to order 14, the elementary excitation gap to order 11 and the 1QP spectral weight to order 10 in the perturbation parameter. Compared to Ref. [47], we were able to add two more perturbative orders to the gap series. Furthermore, it should be stressed that the pCUT approach for long-range interactions was only applied to the 1QP excitation gap so far [47, 50, 51] and is here extended to the ground-state energy and to observables, specifically to the 1QP spectral weight. For a more elaborate discussion on the recent progress of the pCUT approach for long-range systems we refer to the work in progress in Ref. [80].

### 3.3 Full set of critical exponents and the pseudocritical exponent ϙ

We now present the critical exponents of the ferromagnetic LRTFIM for different decay exponents $\sigma \in [0.3, 3]$. This includes the long-range mean field regime $0 < \sigma < 2/3$, the intermediate regime $2/3 < \sigma < 2 - \eta_{\text{SR}}$, as well as the onset of short-range criticality for $\sigma > 2 - \eta_{\text{SR}}$. In addition, we will present the pseudocritical exponent ϙ extracted for different decay exponents in the long-range mean field regime. Before we show the full set of critical exponents, we first present the exponents directly extracted by SSE and pCUT in Fig. 1 and Fig. 2 respectively.

For SSE, those are the exponents $\nu/ϙ$ and $\beta ϙ/\nu$ determined by a data collapse of the mean squared magnetization $\langle m^2 \rangle_L$ as well as $\gamma ϙ/\nu$ determined by a data collapse of the order-parameter

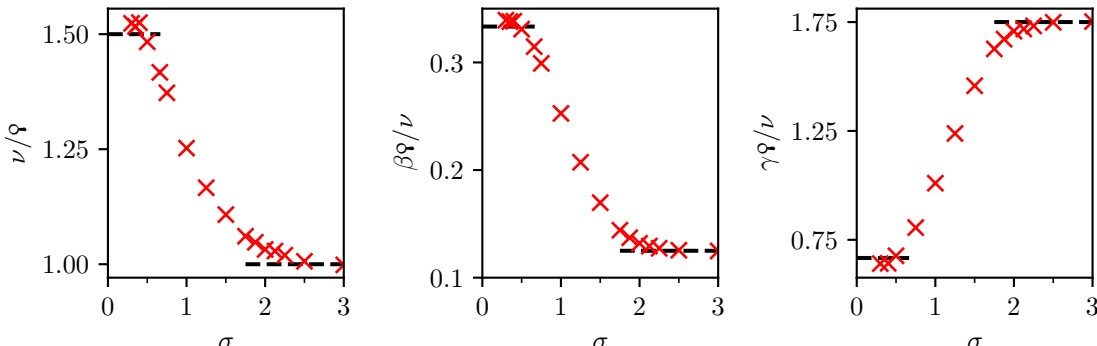

Figure 1: Exponents extracted by data collapses of the SSE data as a function of the decay parameter $\sigma$. The field-theoretical predictions in the long-range mean field regime with $\sigma < 2/3$ (see Tab. 3: $\nu/\varphi = 3/2$, $\beta\varphi/\nu = 1/3$, $\gamma\varphi/\nu = 2/3$) as well as the well-known 2d Ising critical exponents in the short-range regime $\sigma > 2$ (see Tab. 3: $\nu/\varphi = 1$, $\beta\varphi/\nu = 1/2$, $\gamma\varphi/\nu = 1$) are depicted by black dashed lines. The expected shift of the short-range regime to $\sigma > 2-\eta_{\mathrm{SR}}$ [53] is not reflected in the exponents due to a rounding of the boundary. The raw data used to extract the exponents and the numerical values of the exponents are provided in Ref. [71].

susceptibility $\chi_L$. In the long-range mean field regime, the extracted exponents agree well with the field-theoretical predictions with small systematic shifts of $1-2\%$ towards larger values ($\nu/\varphi$ and $\beta\varphi/\nu$) and of $3-4\%$ towards smaller values ($\gamma\varphi/\nu$). This shows that the Q-FSS form Eq. (33) indeed predicts the FSS of those observables correctly. The boundary to the intermediate regime is rounded and at $\sigma = 2/3$, where $d = d_{\mathrm{uc}}$, there is a large shift due to the occurence of multiplicative logarithmic corrections to scaling [81–86] which we did not take into account. In the intermediate regime, the exponents flow monotonously to the well-known critical exponents of the short-range model [87] with the regime boundary between intermediate and short-range regime being rounded. The short-range exponents are in excellent agreement with the ones from the analytic solution [87].

For pCUT, the exponents $zv$, $(2-z-\eta)v$, and $\alpha$ defined in Eq. (68) are determined using (biased) DlogPadé extrapolants from high-order series of the associated quantities. The exponents $zv$ and $(2-z-\eta)v$ are in good agreement with the theoretical predictions in the long-range mean field regime as well as in the nearest-neighbor regime apart from small systematic offsets. The large deviation at the upper critical dimension at $\sigma = 2/3$ arises from the presence of multiplicative logarithmic corrections to the dominant power-law behavior in the vicinity of the critical point [81–86]. For $\sigma < 0.4$, the exponents deviate from the field-theoretical predictions by less than $1.1\%$ for $zv$ and by less than $1.0\%$ for $(2-z-\eta)v$. Further, it should be noted that the spectral-weight exponent resolves the boundary between the intermediate and nearest-neighbor regime better than the gap exponent. The behavior of the exponent $\alpha$ is more subtle as the exponent is expected to be zero everywhere apart from the intermediate regime [25, 87]. In fact, the dominant divergence at the upper-critical dimension as well as in the nearest-neighbor regime is logarithmic which makes the extraction of the exponent demanding. In the nearest-neighbor regime, we account for this logarithmic divergence by using DlogPadé extrapolants biased with

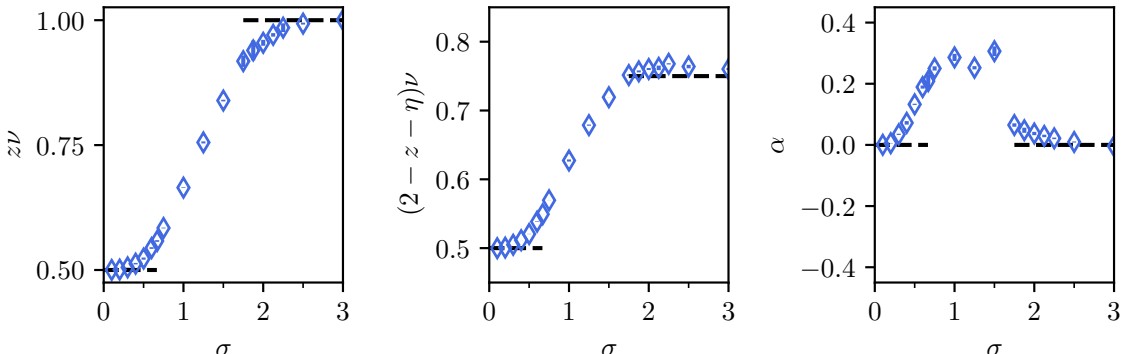

Figure 2: Exponents extracted by (biased) DlogPadé extrapolants of high-order series from pCUT as a function of the decay parameter $\sigma$. The field-theoretical predictions in the long-range mean field regime with $\sigma < 2/3$ (see Tab. 3: $z\nu = 1/2$, $(2-z-\eta)\nu = 1/2$, $\alpha = 0$) as well as the well-known short-range critical exponents in the short-range regime $\sigma > 2-\eta_{\mathrm{SR}}$ [53] (see Tab. 3: $z\nu = 1/2$, $(2-z-\eta)\nu = 3/4$, $\alpha = 0$) are depicted by black dashed lines. The raw data used to extract the exponents and the numerical values of the exponents are provided in Ref. [71].

the expected logarithmic exponent one [88]. However, this results in a jump at the boundary to the intermediate regime where unknown subleading terms to scaling are to be expected that we cannot take into account. Moreover, the deviation from the expected constant mean field exponent for $\sigma < 2/3$ arises due to the presence of the dominant logarithmic divergence at the upper critical dimension $\sigma = 2/3$ [81, 83–85] influencing the extrapolations for $\sigma \leq 2/3$. It should be noted that the direct determination of $\alpha$ is certainly challenging for any numerical technique due to its peculiar behavior.

By means of each method, we extracted a set of three independent critical exponents respectively. The full set of critical exponents was determined using the generalized hyperscaling relation as well as the other common scaling relations. This full set of critical exponents is depicted in Fig. 3. All exponents agree well with the predictions from field-theory in the long-range mean field regime $\sigma < 2/3$ up to the small systematic shifts propagating in the conversion of exponents. The noteable deviation of $\beta$ for small $\sigma$ using the pCUT method can be understood by error propagation of $\alpha$. Analogous, the kinks at $\sigma = 2-\eta_{\mathrm{SR}}$ visible for most of the pCUT exponents come from the methodological artifact of the extraction of $\alpha$ at the boundary between the nearest-neighbor regime and continuously varying exponents. Combining the exponents $z\nu$ and $(2-z-\eta)\nu$ from pCUT with the exponent $\alpha$ from SSE to compensate the deficiency of extracting $\alpha$ directly, the exponents can be further improved in the long-range mean field and intermediate regime. However, the rounding of the exponents at the boundary to the nearest-neighbor regime deteriorates.

The results verify the Q-FSS form Eq. (33) that was used for the data collapse of SSE data as well as the generalized hyperscaling relation Eq. (30) that was used to convert the exponents from both algorithms. Moreover, the study of the long-range mean field regime ensures that the employed algorithms are capable of investigating the demanding regime of long-range interaction with high accuracy. To the best of our knowledge, the full set of critical exponents in the non-trivial intermediate regime is reported for the first time. Previous studies deduced up to two critical

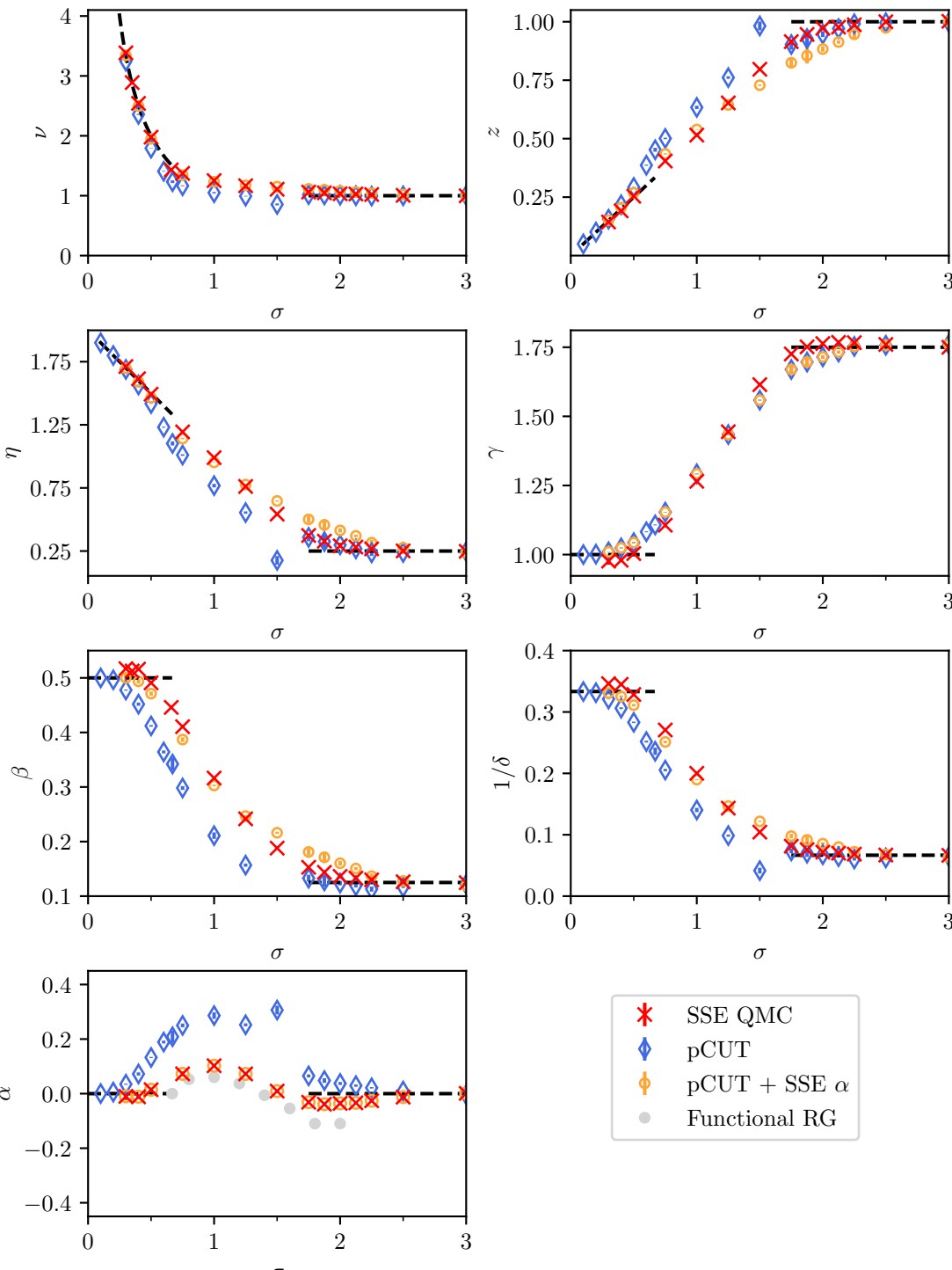

Figure 3: Full set of critical exponents as a function of the decay parameter $\sigma$ extracted by SSE ($\times$), pCUT ($\Diamond$), and pCUT with the $\alpha$ from SSE ($\circ$). The predictions for the long-range mean field and short-range criticality are depicted by black dashed lines for $\sigma < 2/3$ and $\sigma > 2-\eta_{SR}$ respectively. For $\alpha$, we additionally added converted data from functional RG (see Ref. [53], $\bullet$) to compare the bump in the intermediate regime. The raw data used to extract the exponents and the numerical values of the critical exponents are provided in Ref. [71].

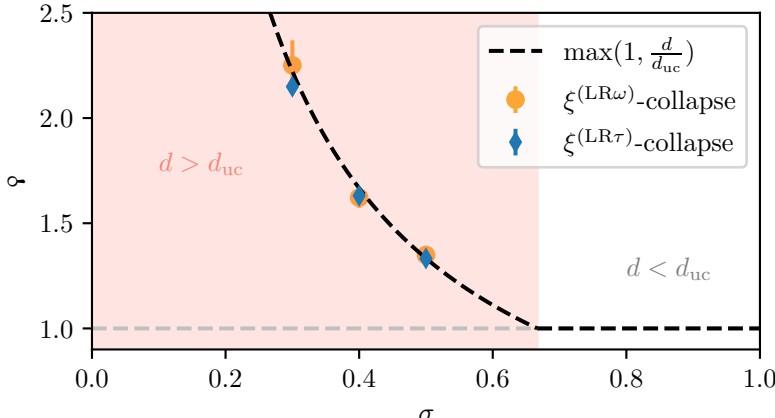

Figure 4: Pseudo-critical exponent $\digamma$ as a function of the decay parameter $\sigma$. $\digamma$ was extracted by data collapses of the finite-size characteristic length scale $\xi_L^{(LR\omega)}$ (see Eq. (56)) and $\xi_L^{(LR\tau)}$ (see Eq. (58)). Both agree within error with the predictions of $\digamma = \frac{d}{d_{uc}}$ in the mean field regime $\sigma < \frac{2}{3}$ and clearly outrule the long-standing belief that $\xi \sim L$. The data of the characteristic length scale used to extract $\digamma$ and the numerical values of $\digamma$ are provided in Ref. [71].

exponents [52–54]. We want to note that the bump in the exponent $\alpha$ for $\sigma \gtrless 2/3$ is not an artifact due to rounding at the boundary and corrections to the dominant power-law behavior close to the critical point, but is also reflected in data of functional RG calculations [53] when converting $\nu$ and $z$ of Ref. [53] to $\alpha$ using hyperscaling. The bump seen in the QMC data when tuning the decay exponent from the short-range regime to the long-range mean field regime is very similar to the "bump" when tuning the dimension of the respective short-range model from $d = 2$ to $d = 4$ with $\alpha = 0$ for $d = 2$ and $d > 4 = d_{uc}$ while, in between, $\alpha = 0.110087(12)$ for $d = 3$ [89]. The remaining exponents interpolate smoothly between the long-range mean field and short-range regimes.

The Q-FSS approach becomes essential in the long-range mean field regime. In Fig. 4, we show the pseudocritical exponent $\digamma$, deduced from the data collapse of the finite-size characteristic length scales Eqs. (56) and (58). Within error, our results agree with the prediction of $\digamma = \max(1, d/d_{uc})$ derived in Sec. 2.2. Our results clearly rule out the long-standing belief that the correlation sector is unaffected by DIV, i. e., $\digamma = 1$ (gray dashed line in Fig. 4).

# 4 Conclusion

We derived a coherent formalism of FSS above the upper critical dimension for continuous quantum phase transitions and confirmed the results numerically by means of the one-dimensional LRTFIM. Our analysis shows that the Q-FSS formalism developed in Ref. [9, 18, 19, 24, 90] for classical systems can be transferred to quantum phase transitions by following our derivation. This provides a tool to extract the critical exponents for continuous quantum phase transitions above the upper critical dimension from finite-system simulations which is especially useful for

unfrustrated long-range interacting systems, where the upper critical dimension is experimentally accessible.

Although the derivation of critical exponents above the upper critical dimension can be easily performed with mean field consideration, we stress the non-trival nature of deriving the same exponents using FSS. The extraction of these critical exponents from finite systems is in and of itself an achievement, which is especially handy to test a method and its accuracy in the context of numerically challenging long-range models as the expected critical exponents are known.

In addition to the possibility of extracting critical exponents, we also introduced a generalized hyperscaling relation. We demonstrated the application of this generalized hyperscaling relation to derive a full set of critical exponents by means of two independent methods with pCUT being a method operating in the thermodynamic limit. This generalized hyperscaling relation makes it possible to perform conversions requiring the hyperscaling relation above the upper critical dimension.

Apart from the mean field critical exponents extracted and converted by Q-FSS, we further present a full set of critical exponents for the one-dimensional LRTFIM in the non-trivial regime of intermediate decay exponents, whereas former studies [52–54] only extracting up to two independent critical exponents.

So far, we applied the formalism only to numerical data, but a certainly formidable application would be to extract critical exponents above the upper critical dimension from experimentally measured observable curves using quantum simulators and applying our approach. This would build a bridge all the way from DIV in the RG flow down to an experimental realization. In this context, we want to note that the boundary conditions play an important role for FSS above the upper critical dimension as the Fourier modes affected by DIV were found to depend on the choice of boundary conditions [9]. For periodic boundary conditions, which are often the first choice in numerical studies but not necessarily in experimental realizations, the zero-momentum observables such as the uniform magnetization are affected by DIV leading to Q-FSS [9], while for free boundary condition other Fourier modes are affected by DIV (see Ref. [9] for details).

Another aspect of our work is the discussion of the connection to the classical Q-FSS theory. Understanding the connection between the classical and quantum case, paves the way to transfer further findings of the classical Q-FSS theory. Possible further studies regarding quantum Q-FSS include an in-depth discussion of the correlation sector in analogy to classical Q-FSS [19], which resulted in an additional $\eta$-like critical exponent $\eta_\mathsf{Q}$ and a corresponding new scaling relation [19]. Furthermore, one could also numerically validate quantum Q-FSS for free boundary conditions where applicable [9] or, on the contrary, validate FSS with the unmodified scaling powers from Gaussian theory for Fourier modes not affected by DIV [9].

Besides systems above the upper critical dimension, there are also other models for which hyperscaling breaks down. In particular, in disordered systems this can also happen due to the appearance of dangerous irrelevant variables [8,91].

## Acknowledgements

We gratefully acknowledge the computational resources and support provided by the HPC group of the Erlangen Regional Computing Center (RRZE).

**Funding information**    A.L., J.A.K., P.A. and K.P.S. acknowledge support by the Deutsche Forschungs-gemeinschaft (DFG, German Research Foundation)—Project-ID 429529648—TRR 306 QuCoLiMa ("Quantum Cooperativity of Light and Matter").

# A    4d nearest-neighbor transverse-field Ising model

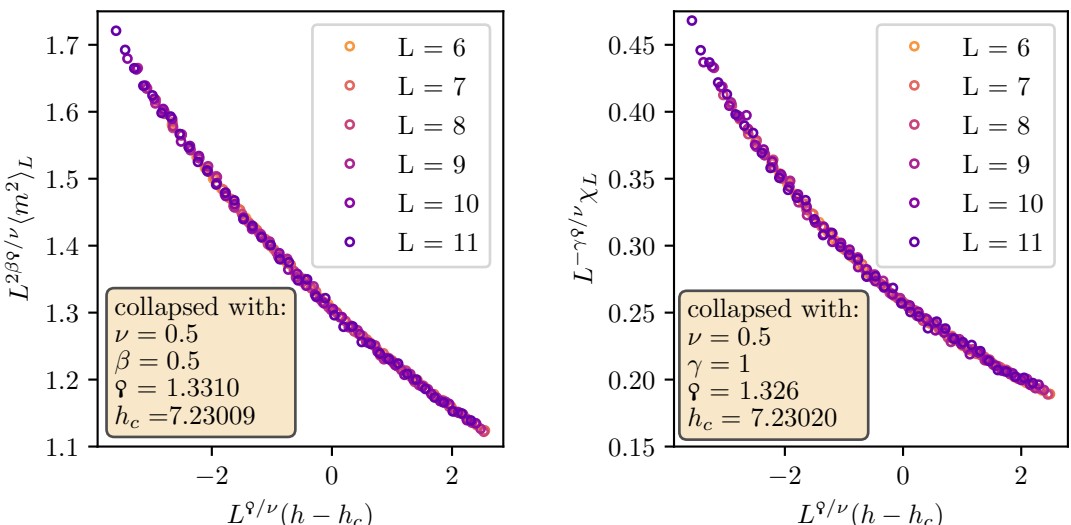

Figure 5: Data collapse of the squared magnetization (left) and the order-parameter susceptibility (right) for the 4d nearest-neighbor TFIM according to Eq. (70). The critical exponents with mean field values $\nu = 1/2$, $\beta = 1/2$ and $\gamma = 1$ were kept fix in the fit. The raw data used for these data collapses is provided in Ref. [71].

In order to draw the connection between classical and quantum Q-FSS, it is instructive to consider the 4d nearest-neighbor TFIM as it corresponds to the 5d classical nearest-neighbor Ising model. The Hamiltonian of the nearest-neighbor TFIM is given by

$$\mathcal{H} = J \sum_{\langle i,j \rangle} \sigma_i^z \sigma_j^z - h \sum_i \sigma_i^x \tag{69}$$

with the sum being restricted to pairs $\langle i,j \rangle$ of nearest-neighbor sites. The universality of this model in $d$ dimensions is the $(d+1)$-dimensional classical Ising universality by virtue of the quantum-classical mapping [62, 92]. The upper critical dimension is $d_{uc} = 3$, making the 4d model the lowest-dimensional representative with $d > d_{uc}$. According to quantum Q-FSS, $\digamma = d/d_{uc} = 4/3$, while for the 5d classical analogue, $\digamma_{cl} = D/D_{uc} = 5/4$. To provide data for this discrepancy, which got explained in Sec. 2.2.4, we perform a data collapse of the squared order parameter $\langle m^2 \rangle_L$ and order-parameter susceptibility $\chi_L$ (see Eq. (33))

$$\begin{aligned}
\left\langle m^2 \right\rangle_L (r) &= L^{-2\beta\digamma/\nu} \mathcal{M}(L^{\digamma/\nu} r), \\
\chi_L(r) &= L^{\gamma\digamma/\nu} \mathcal{X}(L^{\digamma/\nu} r)
\end{aligned} \tag{70}$$

while fixing the mean field critical exponents $\nu = 1/2$, $\beta = 1/2$ and $\gamma = 1$. The collapses of the data is shown in Fig. 5 together with the respective exponents and critical field values. Both fits yield an exponent $\digamma$

$$
\begin{aligned}
\digamma_{m^2} &= 1.3310(9) \\
\digamma_{\chi} &= 1.326(9)
\end{aligned}
\tag{71}
$$

very close to the prediction $\digamma = 4/3$.

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
