# Peer review of "Scaling at quantum phase transitions above the upper critical dimension"

_SciPost Physics_

## Round 1 · Referee Report · Bertrand Berche (Referee 1) · 2022-4-2

Strengths

Quality of the numerical results and of the analysis of the data.
Material presentation of the paper.
Presentation of the theory.
Very reliable conclusions.

Weaknesses

Maybe the QFSS values of the exponents obtained in the plots (values given in dashed lines) could be explained (see my report below), to help the reader.

Report

This paper investigates finite-size-scaling above the upper critical dimension in the quantum Ising model (Ising model in a transverse field) with long-range interactions. Owing to the classical-quantum correspondence, the $d-$dimensional quantum model corresponds to the classical model. For example the critical exponents of the quantum linear chain (with short range interactions, SRI) correspond to those of the Onsager solution of the 2d classical Ising model, e.g. $\beta=1/8$ or $\nu=1$ .
A specific property of the quantum chain is the existence of an infinite ''time direction’’ with a specific dynamic exponent $z$ ($z=1$ for the SRI model). It follows a particular form of the generalized homogeneous functions, e.g.
\[ f(r,H,T)=b^{-(d+z)}f(b^{y_r}r,b^{y_H}H,b^zT)\]
for the free energy density ($r$ is the control parameter, analog to the temperature in classical systems). A consequence of the mapping is also that a finite quantum system in $d$ dimensions corresponds to a classical model with $shape$ $L^d\times\infty$.

Above the upper critical dimension where mean field theory (MFT) exponents are measured, QFSS has been shown to hold, where an exponent ϙ (qoppa) describes the FSS of the correlation length, $ \xi \sim L^ϙ $ with $ϙ = D/D_{uc}$ with $D_{uc} = 4$ for the SRI IM. This exponent modifies the standard FSS relations which become QFSS (with Q for qoppa), e.g. for the magnetization, $m \sim L^{-\beta ϙ /\nu}$. The particular $shape$ of the quantum system changes ϙ into $ϙ = (D-1)/(D_{uc}-1)$. This prediction is confirmed in an excellent manner in the appendix.

For LRI Ising models, $D_{uc} = 2\sigma$ and the MFT exponents also get modified (some of them at least), and QFSS has to be adjusted.
In the case of the LRI quantum Ising chain studied by the authors, $d = 1$, and $z = \sigma / 2$, hence $d_{uc} = 3\sigma / 2$.

The authors present very convincing numerical calculations of the LRI quantum Ising chain for values of $\sigma$ varying from above 2, where the SRI results are recovered (hence the ordinary 2d classical IM) to below $2/3$, the value for which $d_{uc} = 1$, the dimension of the chain. It means that below $\sigma = 2/3$, the 1d system is above its u.c.d and QFSS should hold. The authors present very nice plots of various exponents in the whole range of values of $\sigma$. Their results confirm the expectations, with a crossover between the 2d IM exponents to QFSS. It took me some time to recover the QFSS results, and I think that some details could be given there. For example, at $d_{uc} =d= 1$, $\nu/ϙ = (1/\sigma)/(d_{uc}/d) = 3/2$ or $\beta ϙ /\nu =\beta(d_{uc})\times (d/d_{uc})\times\sigma=\sigma/2= 1/3$. Similar explanations for the other plots might be of interest for the reader.

A very interesting observation made by the authors concerns the choice of the exponents contaminated by the dangerous irrelevant variable above the u.c.d (the starred exponents). Since the constraints appear in ratios, they introduce a free RG dimension $y_L^*$ for the linear length of the system and leave it free. They arrive then at two possible choices, essentially $y_L^* = 1$ (the standard choice made for the study of QFSS) and alternatively $y_r^*=y_r$. This latter choice leads to fix, for arbitrary dimensions above the u.c.d. the RG eigenvalues for the temperature and magnetic field to their Gaussian Fixed Point values exactly at the u.c.d. , $y_r=2$ and $y_H=3$.
This reminds me Coniglio’s picture (A. Coniglio, Shapes, Surfaces and Interfaces in Percolation Clusters, in “Springer Proceedings in Physics, Vol. 5: Physics of Finely Divided Matter”, by M. Daoud, N. Boccara (Eds.), Proc. of Les Houches Cong. on Physics of Finely Divided Matter, Springer Verlag, Berlin (1985) pp 84-101.) of percolation above the upper critical dimension. But there, this picture has been confirmed numerically for systems with free boundary conditions (FBC) only. This is a very particular scenario which leads to the proliferation of interpenetrating spanning clusters, $N\sim L^X$ with $X=D-6$ ($6$ is the u.c.d. for percolation) and for which the RG eigenvalues $y_r$ and $y_H$? remain sticked to their GFP values at $D_{uc}=6$.
This scenario is not observed for periodic systems, where the accepted scenario is compatible with what comes out from $y_L^* = 1$ here (see M. Heydenreich and R. van der Hofstad, Progress in high-dimensional percolation and random graphs, Lecture notes for the CRM-PIMS Summer School in Probability, Preprint (2016). Springer, Cham; Centre de Recherches Math ́ematiques, Montreal, QC (2017).). I think that there are things to investigate there.

The authors present in tables 1 and 2 the values of various quantities for the different choices, $y_L^*=1$ in the first column and $y_r^*=y_r$ in the second, and they give their preference to the second case. My own preference goes to the first column, since it is the one which emerges from Landau theory. Let me give the example of the magnetization, which fixes the value of $p_r$ (I stick to the authors'notations). In Landau theory, the magnetization in the ordered phase varies in $m\sim (-r/u)^{1/2}$ with $u$ the dangerous irrelevant variable. This leads to the GHF, along the lines of Ref [6] (Binder et al)
\[ m(r,H=0,u)=b^{-D+y_H-\frac12y_u}u^{-\frac12}{\cal M}(b^{y_r}r,0)\]
It follows that $\beta_{MFT}=(D-y_H)/y_r+y_u/(2y_r)=\frac 12$ (IM case) and
\[ p_r=-\frac{y_r}{D+y_u}=-\frac 12\]
which is the value obtained in the first column of table 2. As I wrote above, the second column seems to be able to describe Coniglio's picture for percolation (with the appropriate parameters of percolation), so there is more to understand there.

As a conclusion, I find this paper excellent. The above comments are optional and revisions are left to the authors' decision.

Requested changes

The suggestion above on the QFSS limit of the exponents (optional).

---

## Round 1 · Referee Report · Anonymous (Referee 2) · 2022-5-11

Strengths

Deals with an interesting problem and presents solid numerical results
for a very rich model.

Weaknesses

See the report

Report

The hyper-scaling relation and the finite size scaling break down above the upper critical dimension of a model undergoing a continuous transition. This is true for both finite temperature classical transitions and zero-temperature quantum transitions and happens because the interaction term $u$ becomes dangerously irrelevant. In the latter case, the hyper-scaling relation gets modified and involves the dynamical exponent z. Thus the upper critical (spatial) dimension is also changed. In this context, the Q-finite size scaling has been proposed in the context of classical phase transitions. In this work, the same has been generalized here to the quantum critical situations. Generalizing to the quantum case, the authors find that like the classical case, the emergence of a pseudo-critical exponent. This exponent determines the length scale associate with the finite size of the system above the upper critical dimension. Although the pseudo-critical exponent (PCE) has the same form as the classical case, the authors point out a subtle difference. The quantum system at zero temperature is of infinite extent in the imaginary time direction and hence PCE would be similar to that of a classical system having finite size in d (spatial) dimensions and infinity in the temporal direction. The Q-FSS approach proposed in this paper has verified numerically using two numerical methods for a long-range transverse Ising model. In this model, there is a wide parameter range for which mean field theory is valid even when the spatial dimension is unity.

The paper deals with an interesting topic and numerical results are thorough and solid. However, I have following concerns about the work:

a) If one assumes the classical and quantum correspondence, is not the expression for the PCE is immediately obvious? The scaling proposed here is more like a generalization of the already known results. Authors would like to comment on that.

b) Is equation (35) valid for the classical situation also? Below the upper critical dimension, should not $\xi_L \sim L$ always at least for short-range interacting case? The authors should clarify this point here. ( I may be missing something)

c) In 2.2.4, the authors assume that the quantum system is at T=0. If not then the temporal direction will also be finite. Can one generalize the Q-FSS to this most generic situation?

d) If we consider random systems (e.g., classical or quantum random field Ising models) with short-range interactions, the disorder may be dangerously irrelevant and the hyperscaling relation gets modified. Authors may add a note on this.

e) Further, there could be quantum transitions which do not have a classical counterpart; what will be the fate of this scaling in those situations?

I suggest that the authors address these issues more critically, before the paper could be considered for publication.

Requested changes

See the report

---

## Editorial Decision

resubmitted